# Establishing a Reliable Assessment of the Green View Index Based on Image Classification Techniques, Estimation, and a Hypothesis Testing Route

Yiming Liu [1,2], Xiangxiang Pan [2,3], Qing Liu [3,4] and Guicai Li [2,3,*]

1 Guangdong Academy of Social Sciences, Guangzhou 510635, China; liu_yiming@foxmail.com
2 Shenzhen Graduate School, Peking University, Shenzhen 518055, China; 2101212626@stu.pku.edu.cn
3 Laboratory for Urban Future, Peking University, Shenzhen 518055, China; liuq@pkusz.edu.cn
4 Shenzhen New Land Tool Planning & Architectural Design Co., Ltd., Shenzhen 518172, China
* Correspondence: lgcpku@126.com

**Abstract:** Sustainable development policies and spatial planning for maintaining greenery are crucial for all major cities in the world, and the measurement of green space indicators in planning practice needs to evolve in response to the demands of the times and technological drivers. This study explores an informal urban green space indicator, the green view index (*GVI*), which uses the visual perception of an observer to measure the quality of urban space by simulating the pedestrian perspective of the road in street-view image data and then calculating the proportion of vegetation in the road landscape. The *GVI* is different from macro indicators, such as public recreational green space, forest coverage, and green space rate, which are derived from planning data or remote sensing data in traditional urban planning; it starts from the bottom-up perception of individual residents and is more relevant to their subjective demands. At present, most international cities have made outstanding achievements in controlling public recreational green space, forest coverage, green space rates, and other macrolevel indicators of urban spatial quality; however, with the promotion of the concept of "human-oriented" urban planning, the potential restoration of urban spatial quality at the microlevel is gradually being recognized. To ensure the efficiency and reliability of this study, inspired by computer vision techniques and related *GVI* studies, a research method based on chromaticity was built to identify the proportions of green vegetation in street view images, and the credibility was improved by eliminating unreliable data. By using this method, we could evaluate a city at an overall scale instead of the previous block scale. The final research result showed that Shenzhen is friendly to human visual senses, and the *GVI* of the streets in developed areas is generally higher than that in developing areas. The geostatistical analysis of the green viewpoint data provides a more intuitive guide for researchers and planners, and it is believed to inform the planning and design of environmentally friendly, smart, and sustainable future cities.

**Keywords:** the green view index; street-view map; chromaticity; credibility

## 1. Introduction

Urban green space contributes to social and economic benefits, leading to the achievement of sustainable, resilient, inclusive, and competitive urban areas [1], and it has been proven to be essential for improving the health of residents [2–4]. This support includes environmental improvements [5,6], reduced stress [7–9], encouragement for more outdoor activities [10,11], and enhancement of social cohesion and local attachment [12–14]. Urban green space planning generally prioritizes formal and recognized spaces such as parks, forests, and public gardens [15], but often neglects street-level greenery, which is more closely tied to the pedestrian experience. These urban green spaces are highly managed by using officially collected data, which provide the basis for extensive research [16,17]. With the widespread use of street-view image data [18], we believe that green space has

significant new value [19,20], which was not previously noticed by the formal management. Microlevel green views are essential for fostering blocked vitality and further enhancing the overall vitality of a city [21]. Therefore, under the concept of "planning refinement", urban landscape assessment and monitoring guided by observer perceptions are the direction of this paper. The green view index (*GVI*) refers to the proportion of green vegetation in an urban landscape from the perspective of the observers [22]. In this study, we further define the coverage of a street-view image as the proportion of green vegetation in the urban landscape in the streets connecting human living spaces, production spaces, and ecological spaces from the perspective of the observers. Inspired by computer vision technology and related research on *GVI*, this paper uses image classification technology based on deep learning to select street-view image data and establish a reliable assessment of the *GVI*, which can intuitively reflect the perceptions of urban residents. The *GVI* and spatial statistics can further provide policymakers with regional guidelines for urban landscape restoration.

There are already many concepts related to the *GVI*, including public recreational green spaces, forest coverage, and green space rates. Public recreational green space is controlled by urban planning [23–25], and the data often need to be counted, approved, and published by an official government department. The research on forest coverage and green space rates largely benefits from advances in geographic research methods. Remote sensing technology provides reliable, objective, and timely data for both concepts [26–29]. However, public recreational green spaces, forest coverage, and green space rates all come from the macrolevel urban area or from the urban built-up area at the regional level. Therefore, this dimension often has a problem with the applicability of scenes to human groups and individual observers. First, the traditional concept of measuring urban green environmental indicators is not based on the human senses and is not from a humanistic perspective, so it cannot obtain data on human feelings. The *GVI* starts from the human perspective. What humans can see is what the *GVI* uses, so its results are closer to the actual needs of residents. Second, traditional concepts present two-dimensional data results for the planning scope or remote sensing data and are all from an aerial view. The *GVI* uses street-view images to simulate human three-dimensional vision through its parameter settings, so it presents three-dimensional data from the "human view". Third, policy proposals corresponding to the traditional indicators can only have indirect effects on humans. Green spaces' oxygen emissions [30], adsorption of other gases [31], rainwater retention [32], and comprehensive benefits [33] can only be known through professional evaluations. In contrast, policy proposals corresponding to the *GVI* can have direct effects on humans. While urban landscape restoration may be as simple as adding two rows of vegetation on either side of a street to connect human living space, productive space, and ecological space, the enhancement of the visual perception for residents is immediate and obvious. Plants also block ultraviolet light and reduce temperatures close to roads.

Research on the *GVI* began in the field of environmental psychology. It links real scenes with the human visual senses and holds that green plants can stimulate the human brain through the visual senses [34–36]. Next, the correlation coefficients between the human visual senses and feelings were explored with quantitative research methods. During the process of the above research and development, the *GVI* was, therefore, more directly defined as "the proportion of green vegetation in human vision" [33,37]. In the field of urban ecology, the concept related to the *GVI* is "street trees' abundance" [38,39], which affects the microenvironment. Tree-lined corridors consisting of green plants on both sides of roads can significantly improve the outdoor environment during high-temperature periods [40–42], increase air humidity [43], reduce ultraviolet radiation [44,45], reduce air pollution [46–48], and enrich the diversity of bird populations in cities [49,50]. It is generally believed that a block with a *GVI* greater than 25% will provide pedestrians with positive psychological feelings, and a *GVI* greater than 30% will cause satisfaction [13,51]. Compared with previous studies of research concepts, urban landscape restoration based on the *GVI* is better adapted to the actual needs of humans in cities.

The earlier literature indicated that remote sensing techniques were the primary source for determining the percentage of trees in street-view images. However, the mapping and monitoring of street trees by using remote sensing data present various practical application challenges [52]. While satellite image sources are better suited for analyzing clusters of trees, very-high-resolution (VHR) satellite images are often impacted by urban shadows, such as those of power wires and lampposts [53]. Light detection and ranging (LiDAR) and VHR aerial photography datasets are also costly and gathered all at once, making practical applications vulnerable to inaccuracies [54].

Recently, two trends in assessing urban greenery along street networks have gained attention [55]. Firstly, more affordable, detailed, and crowdsourced street-view images have been available, such as Google/Baidu/Tencent street-view maps. Secondly, deep learning techniques have proven to be more effective than other techniques for extracting abstract features and objects from images [56,57]. By calculating the percentage of detected tree canopy cover pixels compared to the total number of pixels in a picture, street-view images were used to measure the green view index (*GVI*) [38]. The *GVI* serves as a proxy for how urban vegetation is perceived by pedestrians and has been applied to an increasing number of cities [58–60]. However, related studies extracted the *GVI* from different paths, lacked realistic-scenario-based tests, and suffered from large errors [61].

To address these issues, it is necessary to use deep learning image classification techniques to process *GVI* data in street-view images and combine them with realistic scenarios in urban planning to construct a relatively complete route for estimation and formulation of a hypothesis. In this study, we improved the *GVI* and provided the case of Shenzhen to reveal the heterogeneity of a mega-city's *GVI*.

## 2. Materials and Methods

### 2.1. Study Area

Shenzhen, which is located in the southern part of Guangdong province, is south of the Tropic of Cancer and is adjacent to the international city of Hong Kong. In the last three decades, Shenzhen has developed from a border town into a modern metropolis. At present, the land area used for urban construction in highly urbanized Shenzhen accounts for 50% of the total urban area [62], which is the highest proportion for all cities on the Chinese mainland. The research area was all of Shenzhen, between 22°24′ N–22°52′ N and 113°43′ E–114°38′ E. The total area was 1997 km$^2$. According to a preliminary analysis of the land-cover data (FROM-GLC10) [63], urban streets are mostly covered by impervious materials, while green vegetation is primarily distributed on the sides of the streets (Figure 1).

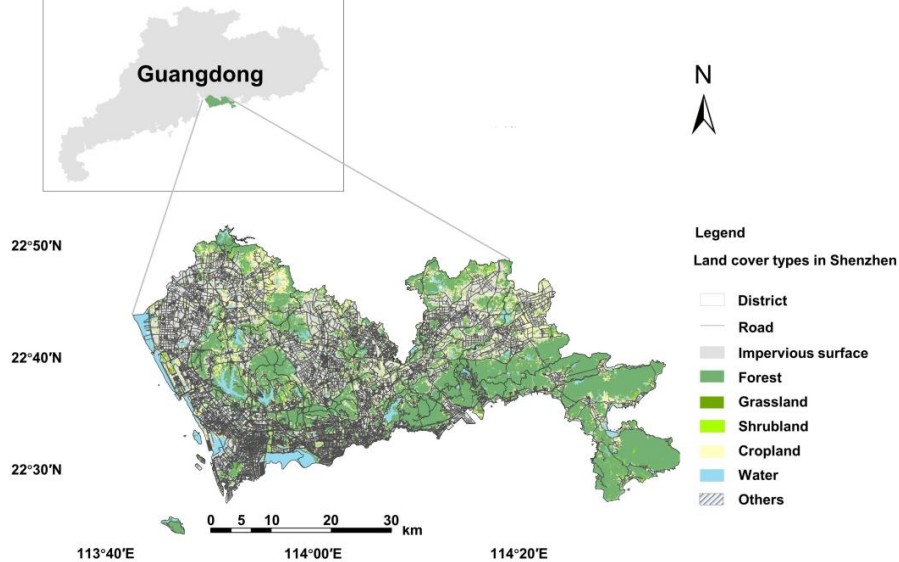

**Figure 1.** Land cover and road layout in Shenzhen, China.

*2.2. Data Source*

2.2.1. Road Data

In this study, the *GVI* came mainly from the range of human activities, so we needed accurate road information. The main road information for Shenzhen came from map data from the open platform Baidu, involving a total road length of 10,068.47 km. The average density of the city's road network is 5.04 km/km$^2$.

The principle for determining random road points was to randomly select 10,068 points (one random point per kilometer, on average) along the main roads of the city by using a GIS, and the distance between the random points was greater than 500 m. The site information (latitude and longitude) was generated from the attributes of the random points, and basic information was established for the next step of scraping the image data.

2.2.2. Image Data

Based on the selection of random road points, we needed to further scrape the image data. To better match the road information and image information, we chose the matching panoramic static map from Baidu.

Based on the longitude and latitude data for the random road points, the image size, the viewing angle parameter, and other information, all required pictures were scraped with a Python program. The viewpoints could be broken down into four images [64,65]: the front image, the back image, the left image, and the right image (Figure 2). To get closer to the pedestrian perspective, we selected pictures that were obtained in four directions (e.g., heading = 0, 90, 180, and 270) from each random point at a horizontal level (pitch = 0) and then adjusted the angle of view to 90° (fov = 90), thus forming a complete pedestrian perspective.

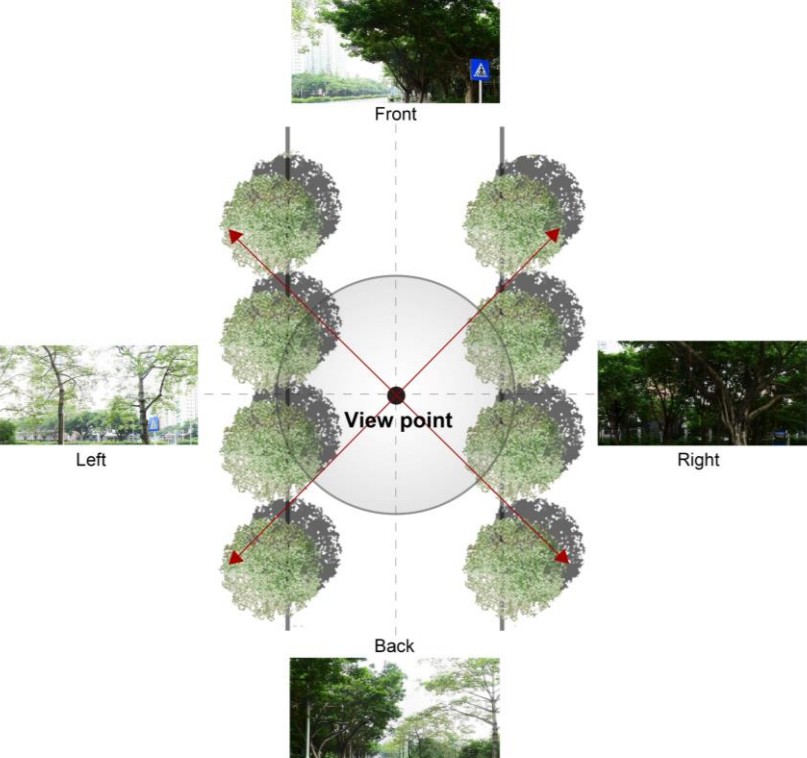

**Figure 2.** The principle of viewpoint picture collection. The urban landscape in a viewpoint can be broken down into four images: the front image, the back image, the left image, and the right image.

2.2.3. Data Processing

In 2006, Hinton and Salakhutdinov used neural networks to reduce data dimensionality [66], which caused deep learning to be reemphasized by academia. In 2017, Krizhevsky,

Sutskever, and Hinton used convolutional neural networks to successfully accomplish visual classification tasks [67], thus making deep learning a mainstream method for computer vision tasks. Presently, deep learning has been widely used in the computer vision field, including in image classification [68], visual scene segmentation [69], and target detection [70]. Computer vision is currently one of the main subjects in computer science. Inspired by computer vision technology and related research on the *GVI*, this paper uses image classification technology based on deep learning to select street-view image data. First, we used the machine to calculate the proportions of green areas in the four images corresponding to each point. Second, we tested the credibility of the green vegetation identifications in the street-view images, one by one. Third, we eliminated unsuitable points by establishing a test model. Finally, we calculated and tested the *GVI* (Figure 3).

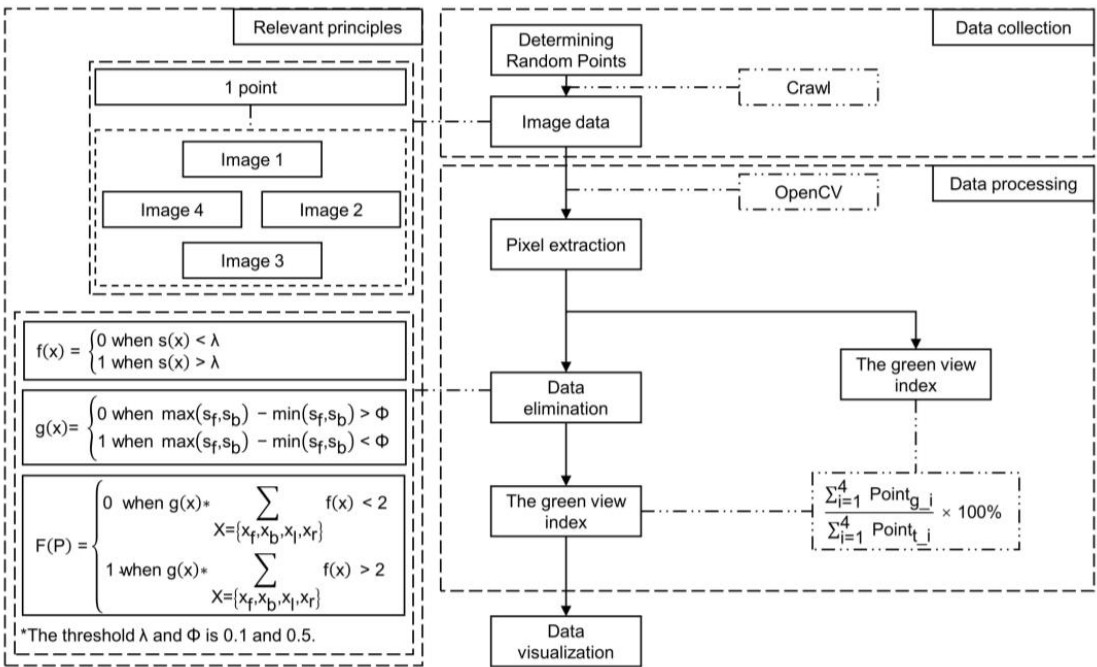

**Figure 3.** Technical route of the research.

(1)     Research Methods

There are two methods for measuring the *GVI* in the computer field. One is to recognize specific colors in images. With the development of computer image recognition, OpenCV tools have gradually become a method for processing images in batches. To make them more conducive for machine recognition, OpenCV tools often transform images from RGB format into HSV format [71]. Another method is to identify features by using scene segmentation and then summing the needed features. The problem with the first method is that the computer often incorporates artificial green items into the calculations, resulting in excessive recognition levels. The problem with the second method is the complexity of defining labels, resulting in insufficient recognition. With the introduction of the concept of "credibility" to general scene recognition in the field of computer image recognition, we can use credibility to restrict the problem of excessive recognition to then ensure the reliability of the recognition results.

By randomly selecting 10,068 points from the main roads of the city, we obtained 26,184 photos from four complete orientations and generated the *GVI* of each picture. Then, the *GVI* values for each picture were connected with the 6546 corresponding points to generate the final result of the *GVI*.

$$GVI = \frac{\sum\limits_{i=1}^{4} Point_{g\_i}}{\sum\limits_{i=1}^{4} Point_{t\_i}} \times 100\% \tag{1}$$

$Point_{g\_i}$ is the number of green pixels in an image taken in direction $i$ of the four directions of an intersection point, and $Point_{t\_i}$ is the total number of pixels in a picture taken in direction $i$ (Table 1).

**Table 1.** Chromaticity analysis.

| General Information | Original Picture | The Picture with the Green View Extracted | Result |
|---|---|---|---|
| |  |  | 0.6474 |
| Location: 114.0632 E, 22.5196 N Picture: width 1024 height 512 |  |  | 0.6330 |
| |  |  | 0.6778 |
| |  |  | 0.5278 |
| | | The viewpoint *GVI* = 0.6215 | |

(2)  Credibility Test

At present, with the advances in deep learning research, researchers have established higher requirements for deep learning accuracy. Therefore, exploring credible deep learning is gradually placed on the agenda. Credibility means making a reliable judgment of the results through recognition. Therefore, the concept of confidence is introduced into deep learning, and it forms a confident deep learning algorithm that provides credibility for each individual image [72]. Rather than using the manual removal of artificial green objects or other features that may present problems, we analyzed the credibility of the green plants in the photos through computer processing and tested all photos one by one, thus providing a reference for the removal of artificial green items.

(3)  Principles of Processing

In this paper, the credibility of image classification expressed the probability that all plants were contained in the detected image. The classification algorithm used sliding windows and multiscale rules, and the classification credibility for image blocks with different locations and sizes was accumulated to ensure that the decision was more accurate. The credibility interval was from 0 to 1. For example, if the credibility of a plant contained in an image was 0.1, then the probability of the plant existing in the image was 10% (Table 2).

**Table 2.** Reliability analysis.

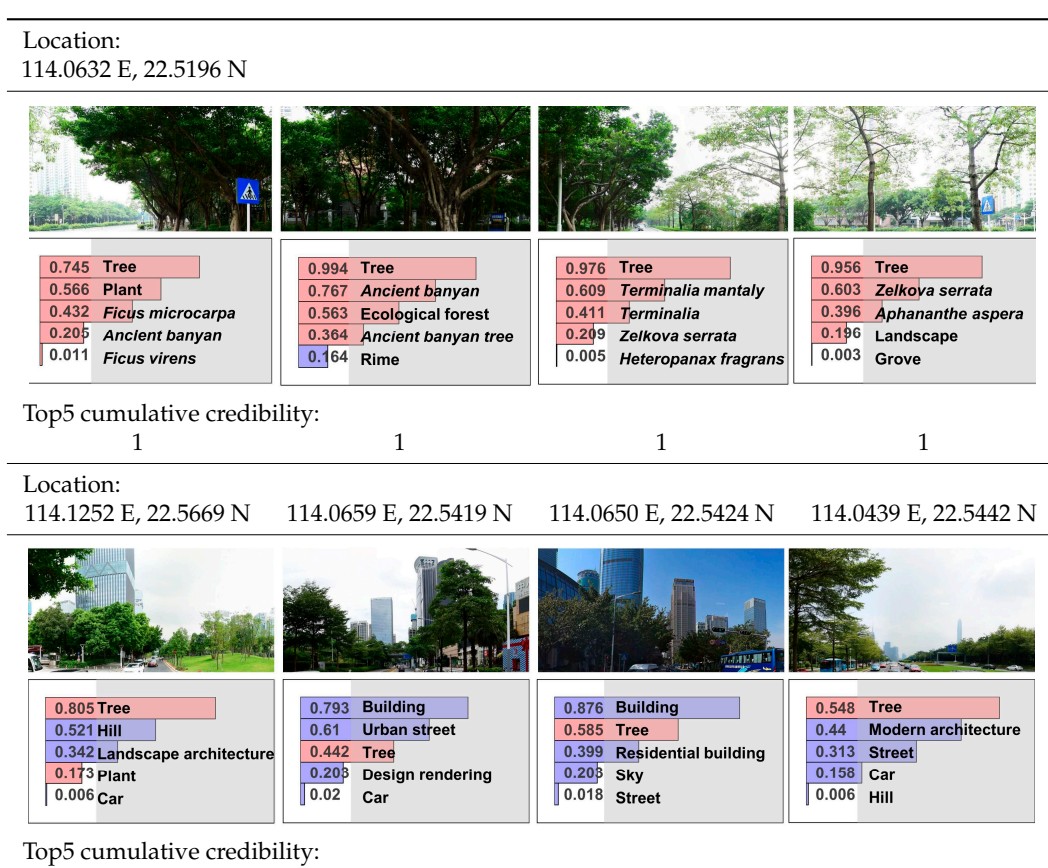

Notes: The credibility interval is from 0 to 1. The pink bars represent plants and the purple bars represent non-plants. According to the top-5 accuracy in image recognition, each image has five key scene recognition values.

We introduced the rejection threshold $\lambda$. When the total credibility (s) for a plant contained in an image $x$ was greater than $\lambda$, the image data were retained, and for the opposite case, the image was rejected. For the formula, $f(x) = 0$ denotes elimination and $f(x) = 1$ denotes reservation.

$$(x) = \begin{cases} 0, \text{ when } s(x) < \lambda \\ 1, \text{ when } s(x) > \lambda \end{cases} \tag{2}$$

For a viewpoint on a road, we collected the image data in four directions—front, back, left, and right—based on one direction facing the road extension direction $X = \{x_f, x_b, x_l, x_r\}$. We input four pictures into the classification algorithm to obtain the credibility $S = \{s_f, s_b, s_l, s_r\}$ of the plant-containing images. For the four pictures of the front, back, left, and right directions of a viewpoint, the probabilities of plants appearing were not equal. In general, the front picture and the back picture (facing the road direction) were more likely to have objects such as roads, sky, and vehicles; the left and right pictures were more likely to contain buildings, plants, pedestrians, and other objects. Therefore, we used different values of $\lambda$ for the image data obtained in different directions ($\lambda = \{\lambda_f, \lambda_b, \lambda_l, \lambda_r\}$) to complete image culling according to the above formula. In this experiment, $\lambda$ was a super-parametric array. Based on the experience with the top-5 accuracy in image recognition [31], top-5 labels are often greater than 0.1. At the same time, combined with the tests in this paper, it was considered that this threshold was in agreement with both the empirical values and the actual research situation. Therefore, this article used $\lambda_f = \lambda_b = 0.1$ and $\lambda_l = \lambda_r = 0.1$.

To avoid errors caused by the image classification algorithm, we proposed the "road constraint" to eliminate the potential errors. According to practical experience, the image data $x_f$ and $x_b$ of a viewpoint have very high similarity for the *GVI*, so there is a positive correlation between the plant-containing credibility of sf and sb. If there is a significant difference between sf and sb, this may indicate an error caused by the image classification algorithm, so we should eliminate this viewpoint P. We introduced the difference threshold $\Phi$ and defined the formula $g(x)$ for the "road constraint". When $\Phi = 0.5$, we believe that there are essential differences in corresponding images, so we set $\Phi = 0.5$ in this study.

$$g(x) = \begin{cases} 0, & \text{when } max(s_f, s_b) - min(s_f, s_b) > \Phi \\ 1, & \text{when } max(s_f, s_b) - min(s_f, s_b) < \Phi \end{cases} \tag{3}$$

Ultimately, whether a viewpoint is rejected depends on the results from the four pictures and whether the "road constraint" is satisfied. That is, a viewpoint will be retained only if the reliability of more than two pictures containing plants exceeds the threshold, and the viewpoint P conforms to the "road constraint" with $X = \{x_f, x_b, x_l, x_r\}$. The complete elimination formula for the viewpoint P is defined as follows:

$$F(P) = \begin{cases} 0, & \text{when } g(x) \times \sum_{X=\{x_f, x_b, x_l, x_r\}} f(x) < 2 \\ 1, & \text{when } g(x) \times \sum_{X=\{x_f, x_b, x_l, x_r\}} f(x) > 2 \end{cases} \tag{4}$$

In existing research, some studies limited image collection to a specific district [73] or part of the city [61,74], the data collection time was over months and years [59], and the number of units of image data was often less than 100,000 [75]. In this study, we obtained 2633 reliable points (10,532 pictures). The results of our study are consistent with those of previous small-scale street studies, such as with the higher green view index observed in the southern part of Longhua District near the city center [76], and we believe that the data are reliable.

## 3. Results

First, at the level of the exploratory analysis for the geostatistical data, one can see the distribution of *GVI* values (Figure 4). Second, based on comparisons of the inverse distance-weighted model and the ordinary Kriging model, we selected the ordinary Kriging model to predict the city's *GVI* distribution (Table 3).

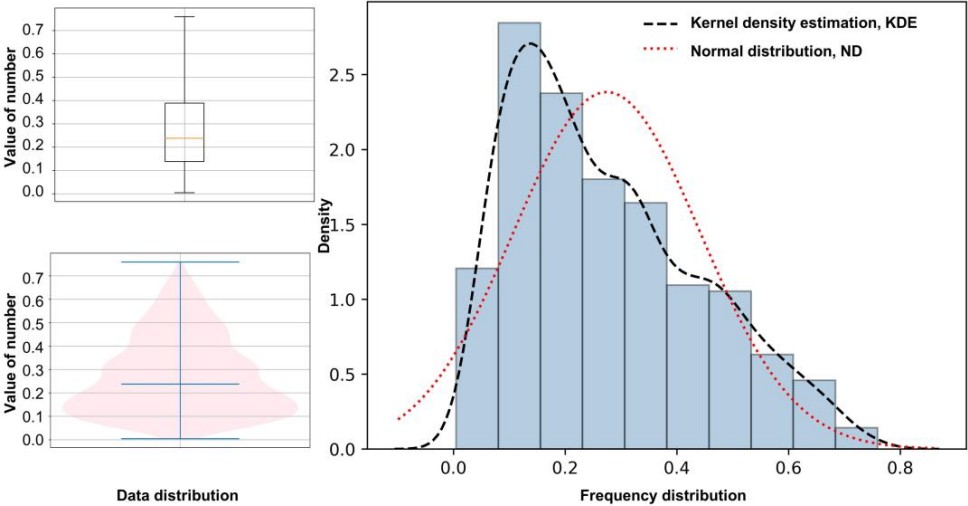

**Figure 4.** Data and frequency distribution: The data distribution clearly shows that the mean is greater than 20% and that the mode is less than 20%; The frequency distribution clearly shows that the distribution of the data is close to an inverted "U" distribution.

**Table 3.** Explanatory model.

| Spatial Interpolation | Inverse Distance-Weighted Model, IDW | Ordinary Kriging Model, OK |
|---|---|---|
| mean | 0.0034 | 0.0007 |
| RMS | 0.1279 | 0.1262 |

Notes: The model selection principle is that the model values of the mean and RMS (root mean square) are both as small as possible.

According to the overall distribution of the data, we can preliminarily see that many areas with high *GVI* values are in southern Shenzhen and that many areas with low *GVI* values are in northern Shenzhen. To further understand the transverse (east–west) and vertical (north–south) properties of the point data, we find that in terms of the transverse relationships of the point data, the *GVI* of central Shenzhen is the highest, that of eastern Shenzhen is the lowest, and that of western Shenzhen is in the middle. From the vertical relationships of the point data, the *GVI* in the south of Shenzhen is the highest, and it gradually decreases while moving northward (Figure 5). At the same time, this study also used the information from a geostatistical analysis that was based on the point data to further estimate the *GVI* distribution for all streets. The average *GVI* in Shenzhen is 28%, and more than half of the city's total urban area has a *GVI* greater than 20%, with some developed areas exceeding 40%. These findings indicate that Shenzhen has a better overall urban spatial pattern in terms of green layout than that of other cities, such as Boston (18.2%), London (12.7%), Los Angeles (15.2%), and Paris (8.8%) [77]. These results show that Shenzhen is friendly to the human visual senses.

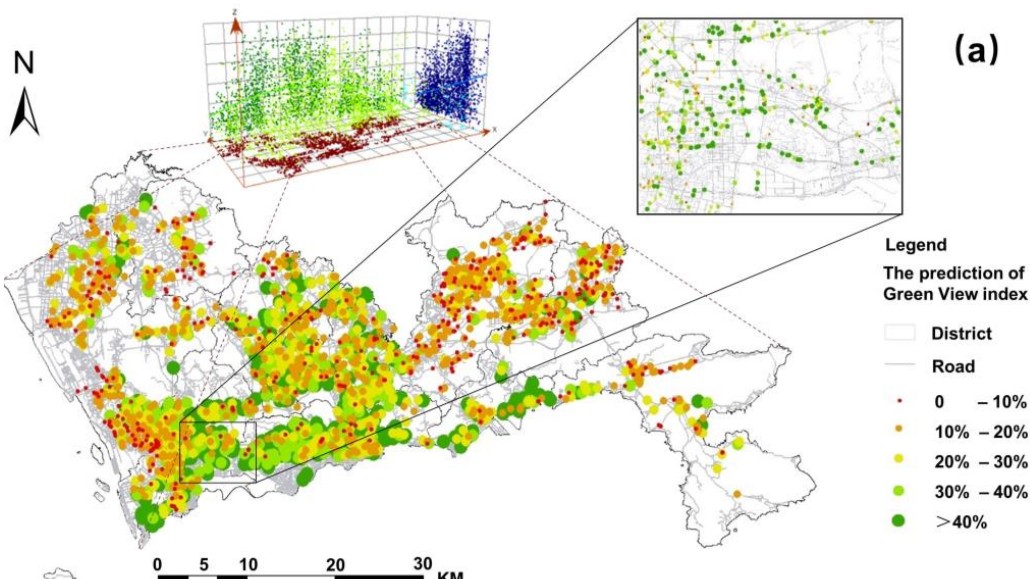

**Figure 5.** *Cont*.

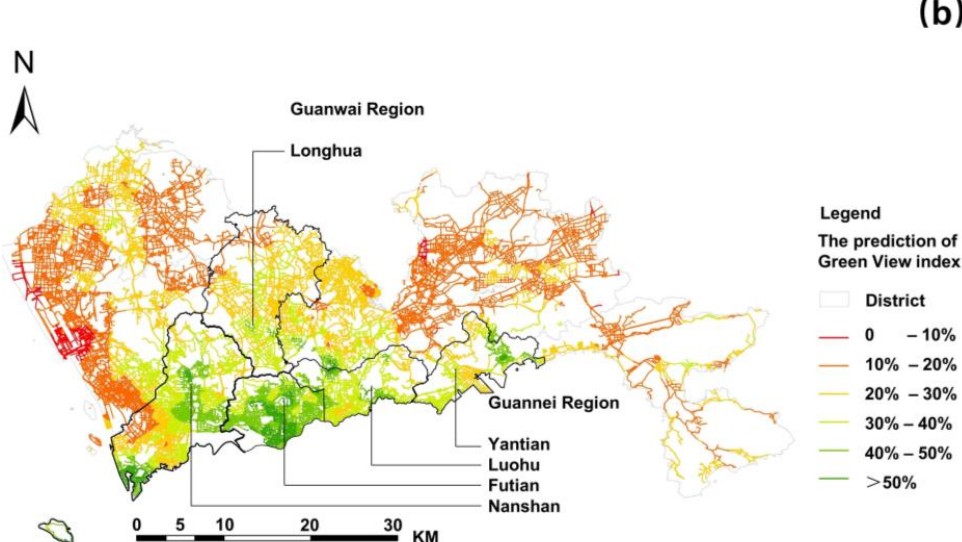

**Figure 5.** The *GVI* in Shenzhen, China. (**a**) Point distribution and (**b**) road layout. As shown by the (**a**) point distribution, the trend in the values can be clearly recognized. In the transverse (east–west) direction, the *GVIs* in the central area are the highest, the *GVIs* for the eastern area are the lowest, and the *GVIs* for the western area are intermediate; for the vertical (north–south) direction, the *GVIs* in the south are the highest and gradually decrease in the northward direction. As shown by (**b**) the road layout, the Shenzhen "Guannei region" is the developed area; the Shenzhen "Guanwai region" is the developing area. The *GVI* values in the "Guannei region" are generally high. In contrast, except for the Longhua district, the *GVI* values in the "Guanwai region" are generally low.

## 4. Discussion

In 1980, Shenzhen was granted the status of a special economic zone, and this was known as the "Guannei region". It was not until 2010 that the border control line within the city was abolished, providing the opportunity for the development of the nearby "Guanwai region". The regions of Guanwai and Guannei have a divided regional economy, with Guannei being a fully urbanized area and Guanwai having a large amount of underutilized and undeveloped land without supportive parks and green spaces.

We preliminarily found that in the southern part of Shenzhen, near Hong Kong, which is the developed area (e.g., Shenzhen "Guannei"), because of the relatively mature urban planning and construction, the *GVI* of the streets is generally high. Compared with "Guannei", the *GVI* of the streets in the eastern, northern, and western areas of Shenzhen (e.g., Shenzhen's "Guanwai", the less-developed areas of Shenzhen) is generally low. Here, the Longhua district, which originally belonged to the "Guanwai" area, has a higher green rate than that of other Guanwai areas, indicating that Shenzhen's "central axis promotion strategy" that was proposed at the macrolevel of urban planning works well. It is worth noting that although the overall *GVI* in Shenzhen is relatively high, there are still a few areas in which it is below 10%, and this must be urgently rectified. From a long-term point of view of the urban landscape, the key areas with a *GVI* of 10–20% need to be improved in terms of the *GVI* of urban roads in Shenzhen in the future. This study considers that these areas are of great significance for improving the overall urban quality of Shenzhen.

## 5. Conclusions

In terms of research content, this study clearly distinguishes the concepts of green environmental indicators in urban images and evaluation systems for urban quality, such as the *GVI*, public recreational green space, forest coverage, and green space rate. Although these indicators can all be applied to urban landscape construction and urban quality improvement, there are differences in their close relationship with human activities and the

efficiency of improvement. Using the *GVI* as a reference indicator for urban planning has a more direct impact on an urban space within the scope of human activities.

In terms of research methodology, this study discusses the problems with the *GVI* measurement methods in previous studies. The first problem is the reliability of the *GVI*'s range, and the second problem is photo rejection. This study used pixel reading as a tool to ensure that all green objects were included in the calculation process, thus avoiding the problem of incomplete labeling. The concept of credibility was introduced to automatically delete images with poor data quality, thus replacing the cumbersome manual elimination process. Therefore, the methods used in this study summarize and extend the research scale, which is advantageous in surpassing previous studies that were conducted on a single-block scale.

In terms of research results, this study suggests that the *GVI* can provide guidance for urban planning. Firstly, the *GVI* represents the quantity of street-level greenery, which provides researchers and planners with intuitive spatial guidance and a decision-making basis for urban optimization. At the same time, the analysis and conclusions based on the *GVI* also reliably reflect residents' personal feelings. It can be used to review the integrity and direction of the layout of urban streets and regional green infrastructure, indicate which areas are visually friendly to residents, and show where more trees should be planted. Secondly, the *GVI*, public recreational green space, forest coverage, and green space rates are different indicators for the planning of urban green space systems, but the goal is to better serve the construction of urban green space. In the future, we can actively explore the effects of linkages between different indicators and, ultimately, build more comprehensive planning.

In addition, this study also has some limitations, such as the lack of detailed consideration of the health of the greenery, the maturity of the trees, the tree species, canopy volume, and canyon geometry in the current methods. Establishing a stable docking mechanism between different street-view images to ensure consistent results for specific locations by using different data sources is also worth considering. Reflecting on these issues will undoubtedly make the application of *GVI* technology more mature and complete.

**Author Contributions:** Conceptualization, Y.L. and G.L.; methodology, X.P. and Y.L.; software, Y.L.; validation, Y.L. and Q.L.; formal analysis, Q.L.; resources, Y.L.; data curation, Y.L. and Q.L.; writing—original draft preparation, Y.L.; Visualization, Y.L. and X.P.; supervision, G.L.; project administration, G.L. All authors have read and agreed to the published version of the manuscript.

**Funding:** This study was funded by Ministry of Science and Technology: National Key R&D Program of China(2018YFD1100802) and Peking University (Shenzhen) Future City Lab: Tiehan Research Fund(202105).

**Data Availability Statement:** Not applicable.

**Conflicts of Interest:** The authors declare no conflict of interest.

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
