# Peer review of "Establishing a Reliable Assessment of the Green View Index Based on Image Classification Techniques, Estimation, and a Hypothesis Testing Route"

_land, doi:10.3390/land12051030_

Round 1

Reviewer 1 Report

The research is interesting and it concerns a contemporary and important issue. There are some interesting findings in the paper but the conclusion is weak. The article deserves to be published. However, in the reviewer’s opinion the article could be improved, if the authors agree, on the basis of the following observations.

In the Abstract (lines 5-7) you mention that “appropriate responses are needed to assess the accessibility of green space”, however accessibility is not addressed in the paper. Perhaps there should be a rephrase.

In your abstract, in lines 20-21 you mention that: “The final research result shows that the average GVI in Shenzhen is 28%”. On its own this observation does not mean anything. It is better that the average of the Guangdong province for example, or compared with some other specific big cities? Maybe you could omit that and mention that ττhe final research result shows that the city is friendly to human visual senses (as you state in lines 276-277) or that the GVI of the streets in the development area is generally higher than the one of the developing area (as you state in lines 290-303) (see also comments below)

Figure 1 has been made by the authors or it is taken from some other source?

In lines 248-250, you mention that based on your research the average GVI in Shenzhen is 28%, which you consider to be reliable compared with the existing research data and the regional locations. It would be useful to elaborate on these “existing research data and the regional locations” so that it can be understood what the value/significance/novelty of the method that you propose is, for example its data are more accurate than existing research data? (a relevant comment could also be added in the abstract, lines 20-21)

In lines 275-276, you mention that the “overall urban spatial pattern of Shenzhen is better than that of other cities in terms of its greening layout”, which cities? Are these cities comparable? Maybe you could add some references.

In lines 276-277 you mention that “These results show that the region is friendly to human visual senses”, you mean the city?

In line 286 you mention that Guannei region “is a prior development area” and in line 291 that is “priority development area”, what do you mean exactly, that is the primary/main area that is developed or that it has a priority by the authorities?

Your finding that the GVI of the streets in the development area is generally higher than the one of the developing area (paragraph 290-303) is really interesting an important. Maybe it should be mentioned in your abstract. It would be useful to example what do you mean by “developing area”, it regards only the built-up area/ construction or also socio-economic factors. Is it possible that the “developing area” does not have mature vegetation/trees?

In sub-section “3.5. Expansion of Research Scale”, you could mention, only if you think is relevant in your research/paper, for some “problems” of your method, for example does this method account for the health of the greenery, the maturity of the trees, or the tree species [given that specific combination of tree species, canopy volume, canyon geometry, and wind speed and direction can actually worsen street-level air quality – which can also be mentioned in your introduction (only if you think that is relevant)]? This could also be incorporated in your conclusions as issues for further research in the future

In lines 330-331, you mention “Thus, this paper argues that the GVI measurements play a guiding role for planning”. In what way GVI measurements can help planning, in finding and acquiring spaces (for example to indicate places for parks), or to indicate in which streets we should plant more trees?  Thus, it can helps spatial and urban planning or landscape architecture, policymakers and Municipality authorities (keeping in mind that GVI represents the quantity of street-level greenery). You can incorporate this issue in your conclusions (see comment below)

You should really elaborate more on your Conclusions. You should really say how “the GVI provides researchers and planners with intuitive spatial guidance and a decision-making basis for urban optimization”. It is not sufficient just to mention that it can help planners/decision-making [you have already mention this in your abstract (lines 21-23), in you introduction (lines 41-42) and in sub-section 3.6 (lines330-331)].

A final remark, in your paper you refer to urban green space [including parks, forests, public gardens (line 33), forest coverage and green space rate, however GVI regards the greenery along the streets, the amounts of greenery experienced by pedestrians. Maybe you should mention that. Or you could mention that there should be a combination with other methods, tools, indicators (if you think that there should be).

Minor mistake in line 32: capital U in urban green space

the quality is ok, minor revisions

Reviewer 2 Report

- Good job on this piece. Thanks for this information-rich paper, where your hard work is very obvious and well structured.

- The knowledge presented here, through quantitative studies, definitely adds to the field of researching greenspaces, especially from the scope of how humans perceives greenery.

- The study is also adding to the existing empirical research on specified areas/cities, and in this case Shenzhen, in Guangdong province, China.

- Please do read all the comments attached to the PDF file very carefully, and improve your draft accordingly.

- one main comment is related to your definition of GVI. It is brief and comes very late in your text (where it should be clarified in the introduction). The reader should not be waiting until the methods sections to see what is GVI.

- Another main comment is your discussion on how this method (as a quantitative measure of greenery - from a human perspective) could be implemented in city/town planning, how it could inform policies. this idea should be further expanded towards the end of your text.

- An additional consideration that could significantly improve your discussion section is comparing the findings of your study (in Shenzhen) to other studies from literature (either in different Chinese cities or other countries). This would highlight how your method is different than others, while illustrating how greenery is integrated in different cities.

- Please make sure to differentiate between what is your own ideas and what you have got from references, especially in the earlier paragraphs of your paper.

Please make sure to use quotations in the correct way in sentences.

Please revise the capitalisation of the first letter of your sentences.

Please revise the grammar of your text.

Reviewer 3 Report

Dear Author(s),

I congratulate you on your magnificent work, the transcendence of your proposal is direct to urban environments. However, I leave you some indications to improve your work:

1. I would improve the quality of the graphics. It is a pity that such important data are seen with poor quality because they are small or somewhat pixelated.

2. I would further develop the results and discussion by implementing them with proposals for future studies that could include automatic or semi-automatic checks, e.g. with Google Street View images.

3. The conclusions need to be developed, they have been rather brief and the importance and transfer to society encourage them to be more concrete and elaborated.

Best regards
